# Kallikrein-kinin blockade in patients with COVID-19 to prevent acute respiratory distress syndrome

**Frank L van de Veerdonk[1]\*, Mihai G Netea[1,2], Marcel van Deuren[1], Jos WM van der Meer[1], Quirijn de Mast[1], Roger J Brüggemann[3], Hans van der Hoeven[4]**

[1]Departments of Internal Medicine, Radboudumc Center for Infectious Diseases (RCI), Radboudumc, Nijmegen, Netherlands; [2]Department for Genomics and Immunoregulation, Life and Medical Sciences Institute (LIMES), University of Bonn, Bonn, Germany; [3]Department of Pharmacy, Radboudumc Center for Infectious Diseases (RCI), Radboudumc, Nijmegen, Netherlands; [4]Intensive Care, Radboudumc Center for Infectious Diseases (RCI), Radboudumc, Nijmegen, Netherlands

**Abstract** COVID-19 patients can present with pulmonary edema early in disease. We propose that this is due to a local vascular problem because of activation of bradykinin 1 receptor (B1R) and B2R on endothelial cells in the lungs. SARS-CoV-2 enters the cell via ACE2 that next to its role in RAAS is needed to inactivate des-Arg9 bradykinin, the potent ligand of the B1R. Without ACE2 acting as a guardian to inactivate the ligands of B1R, the lung environment is prone for local vascular leakage leading to angioedema. Here, we hypothesize that a kinin-dependent local lung angioedema via B1R and eventually B2R is an important feature of COVID-19. We propose that blocking the B2R and inhibiting plasma kallikrein activity might have an ameliorating effect on early disease caused by COVID-19 and might prevent acute respiratory distress syndrome (ARDS). In addition, this pathway might indirectly be responsive to anti-inflammatory agents.

\*For correspondence:
frank.vandeveerdonk@
radboudumc.nl

## Introduction

COVID-19 infects mainly elderly and people with cardiovascular risk, such as hypertension (*Guan et al., 2020*). The clinical spectrum and imaging are so specific that MDs recognize this disease in an instant especially now that it is widespread. Every clinician recognizes that the virus does not cause disease similar to influenza, which carries the risk that designing targeted therapies based on the pathogenesis of influenza might fail in COVID-19. Research on Sars-CoV pathogenesis which might be very similar to Sars-CoV-2 pathogenesis has focused the discussion on ACE inhibitors, recombinant ACE2, and ARBs and how they could fit in the pathogenesis of COVID-19, since these pathways were extensively studied in SARS (*Fang et al., 2020*; *Batlle et al., 2020*). For recombinant ACE2 this would be straight forward, it would at least be an attempt to bind and try to scavenge the virus (*Batlle et al., 2020*). However, for ACE inhibitors and ARBs it is a much more complicated story. Since most of the attention was focused on the RAAS system and its interaction with modulating the vascular system and inflammation, the other major role of ACE and ACE2 for the regulation of the kinin-kallikrein system was lacking attention (*Jurado-Palomo and Caballero, 2017*; *Marceau et al., 2018*). Moreover, the notable clinical deterioration seems to be associated with increased inflammatory status. Here, we describe the clinical observations that brought the clues for explaining the potential pathophysiological mechanisms, and offer a rationale for targeted treatment at different stages of COVID-19.

**eLife digest** The COVID-19 pandemic represents an unprecedented threat to global health. Millions of cases have been confirmed around the world, and hundreds of thousands of people have lost their lives. Common symptoms include a fever and persistent cough and COVID-19 patients also often experience an excess of fluid in the lungs, which makes it difficult to breathe. In some cases, this develops into a life-threatening condition whereby the lungs cannot provide the body's vital organs with enough oxygen.

The SARS-CoV-2 virus, which causes COVID-19, enters the lining of the lungs via an enzyme called the ACE2 receptor, which is present on the outer surface of the lungs' cells. The related coronavirus that was responsible for the SARS outbreak in the early 2000s also needs the ACE2 receptor to enter the cells of the lungs. In SARS, the levels of ACE2 in the lung decline during the infection.

Studies with mice have previously revealed that a shortage of ACE2 leads to increased levels of a hormone called angiotensin II, which regulates blood pressure. As a result, much attention has turned to the potential link between this hormone system in relation to COVID-19. However, other mouse studies have shown that ACE2 protects against a build-up of fluid in the lungs caused by a different molecule made by the body. This molecule, which is actually a small fragment of a protein, lowers blood pressure and causes fluid to leak out of blood vessels. It belongs to a family of molecules known as kinins, and ACE2 is known to inactivate certain kinins.

This led van de Veerdonk et al. to propose that the excess of fluid in the lungs seen in COVID-19 patients may be because kinins are not being neutralized due to the shortage of the ACE2 receptor. This had not been hypothesized before, even though the mechanism could be the same in SARS which has been researched for the past 17 years. If this hypothesis is correct, it would mean that directly inhibiting the receptor for the kinins (or the proteins that they come from) may be the only way to stop fluid leaking into the lungs of COVID-19 patients in the early stage of disease.

This hypothesis is unproven, and more work is needed to see if it is clinically relevant. If that work provides a proof of concept, it means that existing treatments and registered drugs could potentially help patients with COVID-19, by preventing the need for mechanical ventilation and saving many lives.

## Clinical observations

When patients are admitted with symptomatic COVID-19 infection fever, dry cough, and dyspnea are most commonly observed. Importantly, we observed that dyspnea and tachypnea can differ from hour to hour and a feeling of drowning is described with sometimes sudden recovery by patients. CT scans reveals unilateral or bilateral ground-glass opacities, that might progress to more clear consolidations throughout the disease. Fluid restriction improves oxygenation and ameliorates the feeling of dyspnea. Notably, plasma concentrations of D-dimers at this stage are increased without evidence of thromboembolic events. There is a phase during clinical admission where many patients are getting better, but some will worsen especially around day 9, although this can also occur much earlier. This worsening seems to be accompanied specifically with further increases in IL-6, CRP, ferritin, without elevated procalcitonin, indicative of a progressive innate inflammatory status, which is a clear different pattern of the first stage of the disease.

In the ICU, there are several striking observations. In contrast to patients with common forms of ARDS, approximately 70% of patients with severe COVID-19 show an only slightly decreased pulmonary compliance (L-type) (*Gattinoni et al., 2020a*). Driving pressure is usually low. Recruitability is usually low and the use of high PEEP may therefore substantially increase functional residual capacity resulting in hyperinflation, high strain and considerable hypercapnia through an increase in dead space ventilation. Hereby mechanical ventilation may further contribute to lung damage. Only a minority of patients initially show the classical ARDS mechanical properties (H-type) with low compliance, high driving pressure and higher recruitability. Both L and H-type show high venous admixture. We and others have suggested that the L-type may progress to the H-type by a combination of negative intrathoracic pressure and increased lung permeability due to inflammation (so called patient-self inflicted lung injury P-SILI) (*Gattinoni et al., 2020b*).

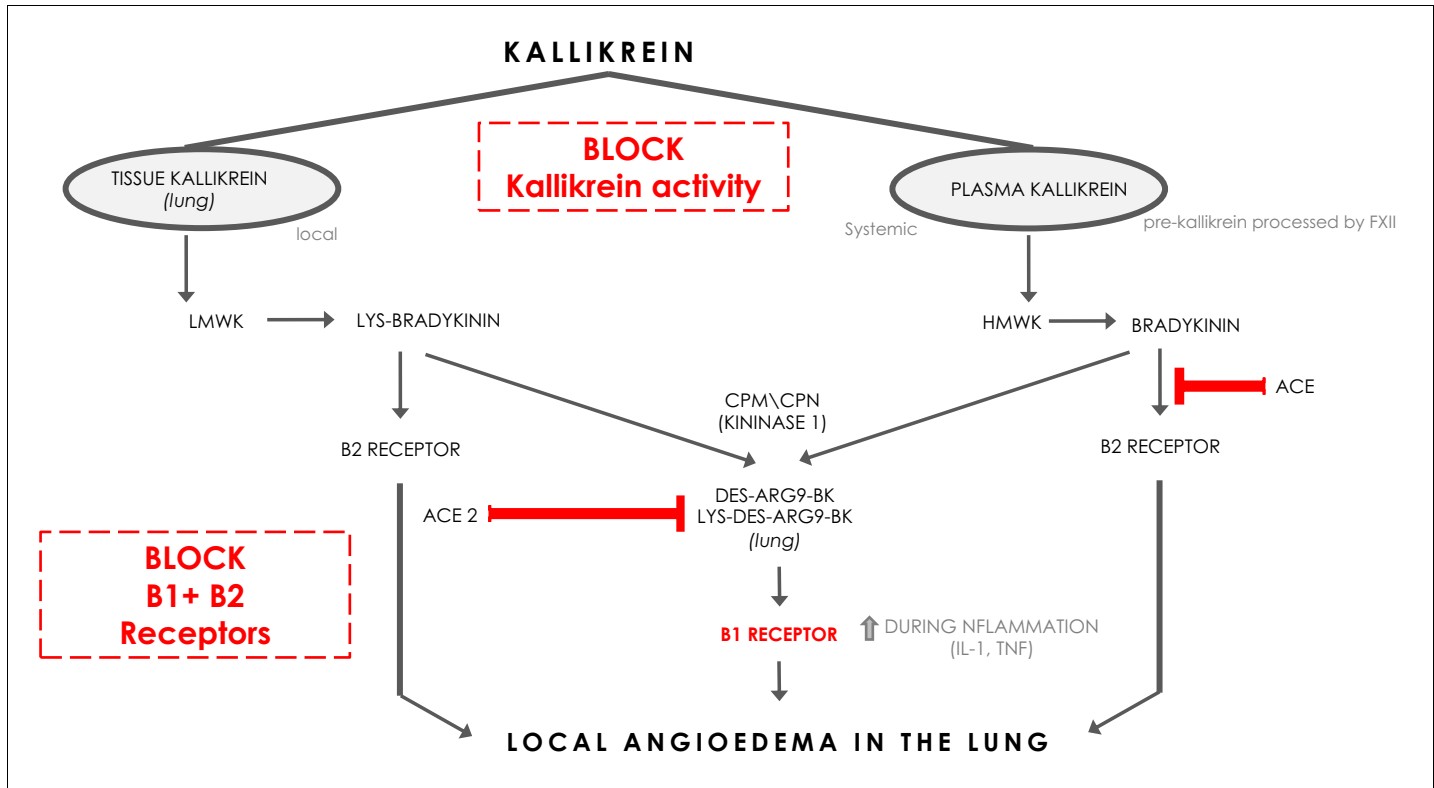

**Figure 1.** The kinin-kallikrein system and ACE/ACE2. The pathways of processing of low-molecular-weight kininogen (LMWK) and high-molecular-weight kallikrein (HMWK) leading to Bradykinin 1 (B1) receptor agonists and Bradykinin 2 (B2) receptor agonists. CPM = carboxypeptidaseM; CPN = carboxypeptidaseN.

## Bradykinin-induced local pulmonary angioedema

We propose it all starts with ACE2 and its role in the kallikrein-kinin system, which to date has not been investigated in the pathogenesis of SARS or COVID-19. The kinin-kallikrein system is a zymogen system that after activation leads to the release of the nona-petide bradykinin that after binding to the B2R on endothelial cells can lead to capillary leakage and thus angioedema. The prototype diseases of local peripheral transient increased bradykinin release are hereditary or acquired angioedema (*Jurado-Palomo and Caballero, 2017*). The clinical picture of COVID-19 is in line with a single-organ failure of the lung that is due to edema at the site of inflammation. Moreover, the presence of an elevated D-dimer without thrombosis or microangiopathy is in line with the high D-dimers in angioedema. This most likely reflects the leakage of plasma substances involved in the coagulation cascade leading to fibrin and due to kallikrein activity is processed into D-dimer and leaks back into the circulation, reflecting subendothelial activation and kallikrein activity. The ACE2 and its role in the RAAS system has been suggested to play a role for more than 10 years in the pulmonary edema due to ARDS and SARS (*Imai et al., 2005*). Pulmonary edema by ACE2 dysfunction was speculated to be due to increased hydrostatic pressure as a result of vasoconstriction of the pulmonary vasculature due to high angiotensin II (a vasoconstrictor) (*Imai et al., 2005*). However, further experiments showed no difference in hydrostatic pressure and made the explanation of high angiotensin II with vasocontriction as a cause of pulmonary edema unlikely (*Imai et al., 2005*; *Kuba et al., 2005*). Increased bradykinin, however, could explain this observation without increased hydrostatic pressure. Notably, the RAAS system controls vasoconstriction and vasodilatation, and the bradykinin system controls permeability and vasodilatation, whereas ACE2 regulates both.

Bradykinin (BK) is a linear nonapeptide that is formed by the proteolytic activity of kallikrein on kininogens (*Bhoola et al., 1992*). Kallikreins are serine proteases and can be divided in plasma kallikrein and tissue kallikreins (*Figure 1*). The plasma and tissue kallikreins release the vasoactive

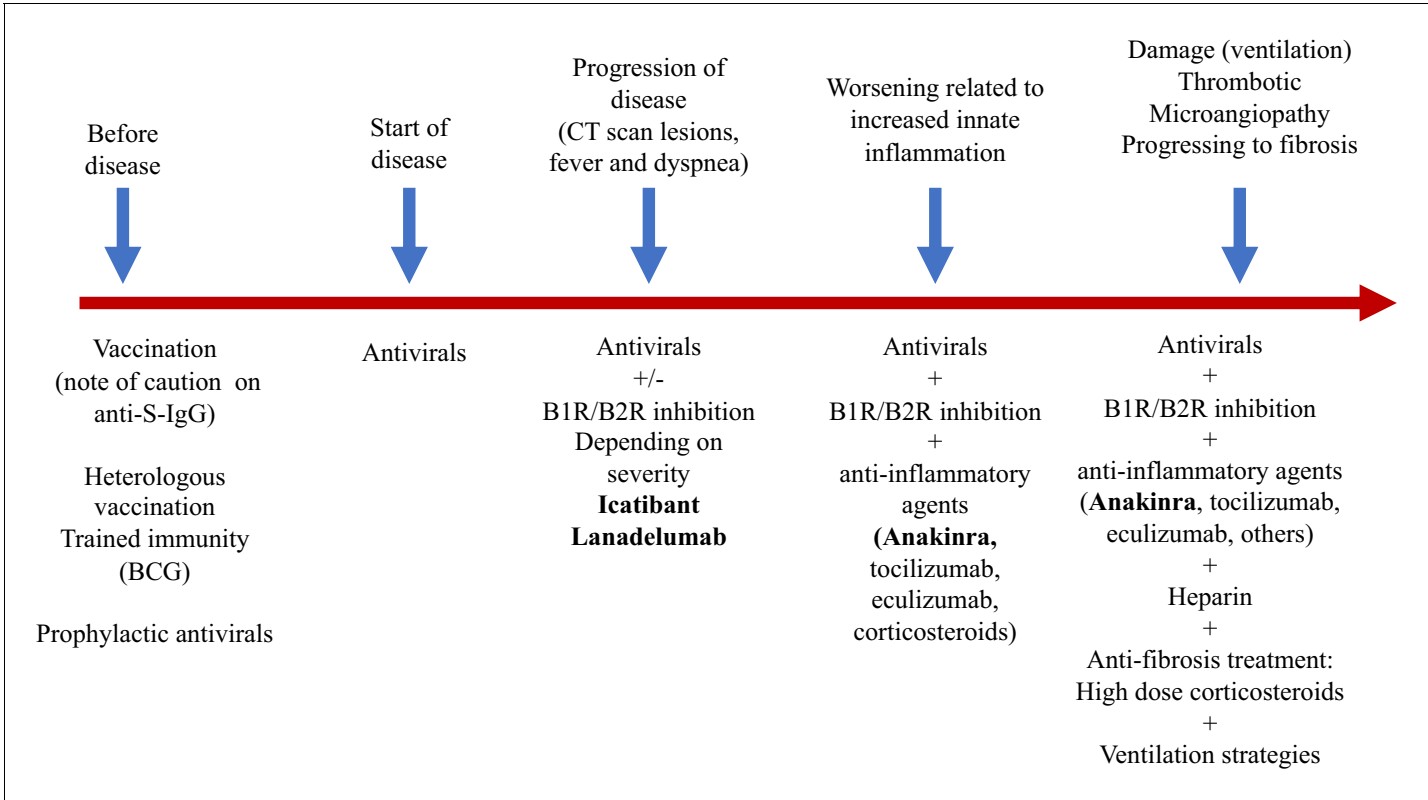

**Figure 2.** Schematic view of treatment strategies in COVID-19.

peptides known as kinins (all sorts of BKs) that cause relaxation of vascular smooth muscle and increased vascular permeability (*Bhoola et al., 1992*; *Marceau et al., 2018*). Plasma kallikrein processes high-molecular-weight kininogen (HMWK produced by the liver *Bhoola et al., 1992*) into bradykinin (BK), while tissue kallikrein processes low-molecular-weight kininogen (LMWK produced by the liver *Bhoola et al., 1992*) and results in Lys-BK (*Figure 1*). These are the ligands for the constitutively expressed bradykinin receptor B2 on endothelial cells (*Jurado-Palomo and Caballero, 2017*). In addition, the enzymes (carboxypeptidase M (CPM) and carboxypeptidase N (CPN)) can further process BK and Lys-BK into des-Arg$^9$-BK and Lys- des-Arg$^9$-BK respectively, which are ligands for B1R, a bradykinin receptor on endothelial cells that is upregulated under proinflammatory conditions (*Jurado-Palomo and Caballero, 2017*). These kinins have strong vasopermeable and vasodilatory capacity and need to be tightly controlled to prevent excessive angioedema. ACE and ACE2 both have roles in inactivating the ligands for the bradykinin receptors (*Gralinski et al., 2018*). ACE mainly inactivates bradykinin which is the major ligand for B2Rs. ACE inhibition has been linked to systemic acquired angioedema since it can result in excessive presence of bradykinin that activates B2R (*Jurado-Palomo and Caballero, 2017*).

Interestingly, ACE2 does not inactivate bradykinin, but can inactivate Lys des-Arg$^9$-BK and des-Arg$^9$-BK which are potent ligands of the B1R in the lung (*Figure 1*; *Sodhi et al., 2018*). In this way, it can be protective against pulmonary edema especially in the setting of inflammation, which is further supported by the role of ACE2 in acute pulmonary injury (*Imai et al., 2005*; *Sodhi et al., 2018*). When plasma leakage occurs due to tissue damage and tissue kallikrein activation in the setting of innate inflammation, plasma kallikrein will be activated locally resulting in the formation of bradykinin that stimulates the B2R and des-Arg$^9$-BK that will further stimulate the B1R. ACE2 is almost undetectable in serum, but is expressed in the lung predominantly on pneumocytes type II (*Sodhi et al., 2018*). The Sars-CoV-2 Spike (S) antigen binds to ACE2 and internalizes (*Walls et al., 2020*). Since it has been reported and suggested that the expression of ACE2 and its capacity of enzyme activity is decreased in SARS-CoV and inflammatory conditions (*Sodhi et al., 2018*; *Imai et al., 2005*; *Kuba et al., 2005*), it is tempting to speculate that Sars-CoV-2 interaction with ACE2 at the surface

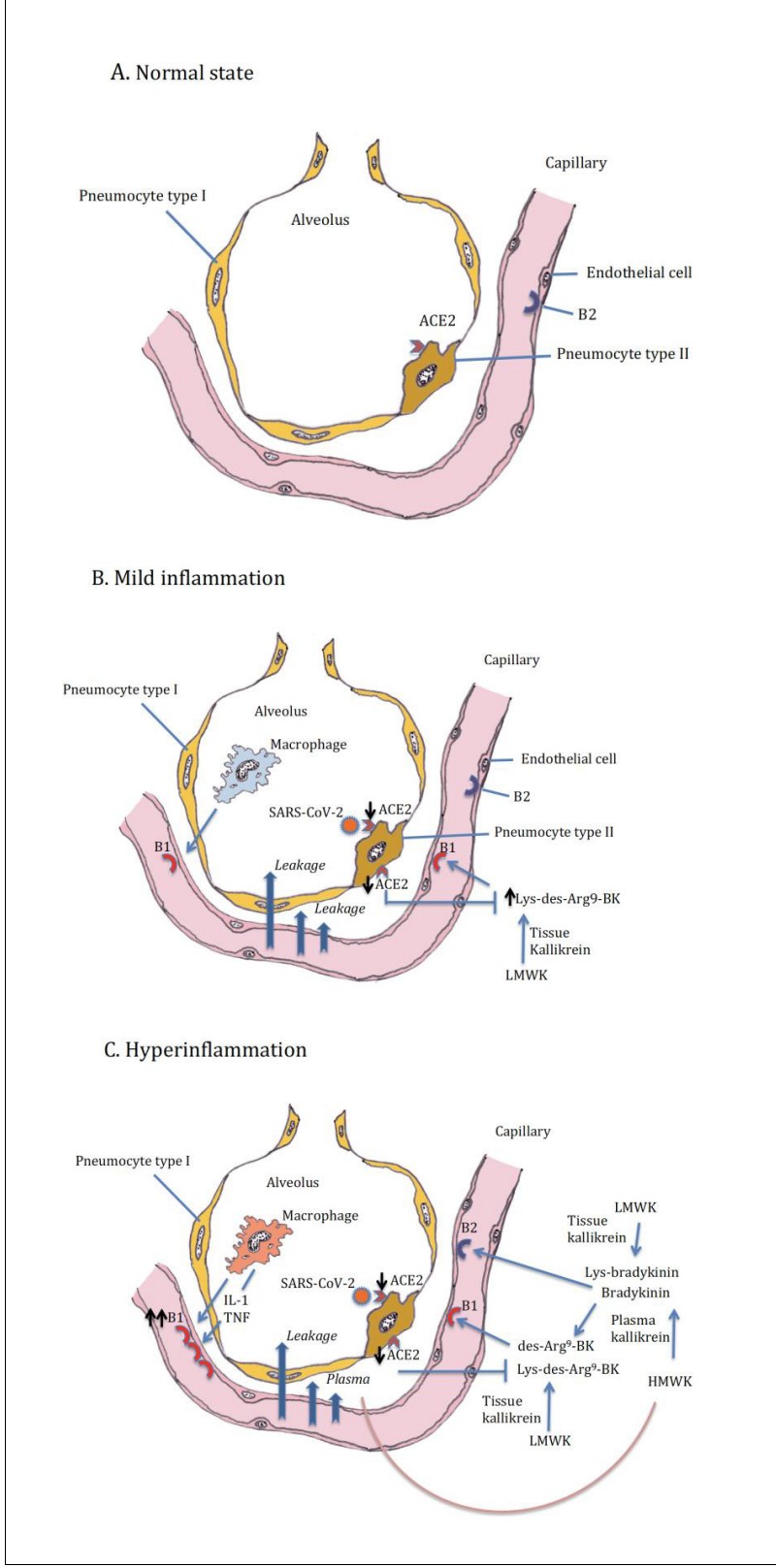

**Figure 3.** Alveolus in normal setting and during moderate and severe COVID-19, (**A**) normal, (**B**) mild inflammation, (**C**) hyperinflammation. ACE2 downregulation by the SARS-CoV-2 is followed by loss of neutralizing capacity of Lys-des-arg$^9$-bradykinin (BK) in the lung leading to plasma leakage. Subsequently plasma leakage results in more B1R ligands (des-arg$^9$-BK) and B2R ligands (bradykinin).

also downregulates ACE2 expression and function of ACE2, subsequently leading to a deficiency to inactivate the B1R ligand locally in the lung, and might in this way directly link the virus to local pulmonary angioedema. Further supporting this concept are the reported findings of downregulation of ACE2 by SARS-CoV, and it has been suggested that this might be similar in SARS-CoV-2 (*Fu et al., 2020*; *Glowacka et al., 2010*; *Levi et al., 2019*).

In 2005, it was proposed that the RAAS system was responsible for complications due to Sars-CoV. RAAS regulates vasodilatation and vasoconstriction, and it was hypothesized that increased angiotensin II as a result of ACE2 deficiency would result in pulmonary edema due to increased hydrostatic pressure since angiotensin II would cause vasoconstriction. However, there was no effect observed on the hemodynamics of the pulmonary vasculature in ACE2 deficiency, while there was clear vascular leakage. AT1R knockout mice and AT1R blockade were protected from lung edema due to inflammation but this was not explained by a mechanism linking AT1R to vascular leakage. Bradykinin might be the missing link, since AT1R can form heterodimers with the B2R and AT1R can synergize with B1R in the induction of ROS in endothelial cells (*Ceravolo et al., 2014*; *Quitterer and AbdAlla, 2014*).

We speculate that this dysregulated kinin pathway is present already early in COVID-19 disease. Patients can worsen clinically after days of illness which is accompanied by an increase in proinflammatory status often resulting in ICU admission and with necessity of supportive mechanical ventilation. Especially a strong innate immune response reflected by high levels of IL-6 and CRP seem to accompany this clinical worsening. This will not only result in more damage to the environment with neutrophil recruitment but will also further increase inflammation-induced B1R upregulation on endothelial cells especially via IL-1. However, it must be kept in mind that targeting the innate immune response will not have a direct effect on the pulmonary edema that is driven by bradykinin, since kallikrein activity will be not affected, kinins will still be present, and B1R and B2R are still expressed on endothelial cells. This pathway might be less responsive to corticosteroids or adrenaline, meaning as long as the virus persists ACE2 dysfunction is present and the bradykinin pathway is active the pulmonary edema at the site of infection will persist. On the other hand, clinicians know how fast patients with bradykinin-related angioedema can recover with for example icatibant or when the trigger is gone that one can foresee a very fast recovery of pulmonary edema and recovery of hypoxia and disease when intervening with the plasma kallikrein-kinin pathway.

## Targeted treatment and timing of interventions

### Blocking B1R and B2R

In our vision, as long as the virus persists the dysregulated kinin-kallikrein pathway is playing a role in disease via the absence of optimal ACE2 function in the lung. Maybe not everybody needs kallikrein-kinin blocking since they will recover once the viral load is resolved from the lung and there is no second inflammatory hit. However, when disease progresses which is accompanied by increased proinflammatory status which often results in critical illness we would argue that this timepoint has a rationale for strategies targeting the inflammation induced by innate immune responses. However this must be done in the presence of blocking the kallikrein-kinin pathway. Several targets might be amendable to intervention, namely 1. at the level of blocking tissue and or plasma kallikrein activity and thus reducing the production of kinins, 2. activating the degradation of kinins by treating with recombinant active enzymes such as ACE2, 3. at the level of B1R and B2R, 4. by inhibiting the common downstream signaling of B1Rand B2R, and 5. by suppressing local NO which is largely responsible for the endothelial leakage.

By far, the most potent and logical would be to block B1R and B2R signaling. B2R inhibitors exist in the clinics. Icatibant is a selective B2R drug that is available in the US and Europe (Firazyr) and is licensed for the treatment of hereditary angioedema in adults, adolescents and children over the age of 2 years. It is a synthetic decapeptide with a structure similar to bradykinin, but with five non-proteinogenic amino acids (*European Medicines Agency, 2014*). The licensed dose of icatibant for hereditary angioedema is 30 mg by subcutaneous injection as a single dose. At current day, there is no licensed B1R drug (*Qadri and Bader, 2018*). Several B1R drugs have been tested in pre-clinical and in phase I/II trials as therapeutic target for inflammation related processes already since the 1970s. None of these drugs have made it to the market. This includes drugs like the Merck

compound MK-0686 (*Kuduk et al., 2007*) that has been investigated in the reduction of pain, and the Sanofi compound safotibant that was discontinued in 2012 for the treatment of macular oedema and the Boehringer Ingelheim drug BI11382 (*Nasseri et al., 2015*). Other products identified via open target (accessible via http://www.opentarget.com) and through literature review are ELN-441958, SSR 240612, NVP-SAA164 and R-715 (*Qadri and Bader, 2018*). Dual inhibition of both the B1R and B2R would be the way forward. But this would imply that specifically the drugs targeting B1R need to be become available and that they have to exert suitable pharmacodynamic action at concentrations that are non-toxic. Another option is the use of blocking plasma kallikrein, which in turn will result in less kinins (both B2R and B1R ligands) at the site of infection and subsequently less leakage via B1R and B2R.

## Antiinflammatory strategies

In addition, we should think about blocking innate cytokines that upregulate B1R on endothelial cells at the site of inflammation in combination with B1R and or B2R blockade. IL-1 (consisting of IL-1α and IL-1β) and TNF are potent inducers of B1R. Blockade of NF-κB translocation, TNF-α, or IL-1 prevented the functional and molecular up-regulation of B1R by LPS (*Passos et al., 2004*). Therefore, one strategy could be with anakinra, which has an excellent safety profile and would make a lot of sense since it not only blocks IL-1β coming from infiltrating monocytes and macrophages, but also IL-1α. IL-1α is likely to be play a role locally due to its release from inflamed endothelial cells. Blocking TNF is an option, but has been associated with much more infectious complications. In addition, complement activation has been described and could play a role in this stage of disease, and this might be amendable to C5 blockade with eculizimab with which a randomized trial in COVID-19 is being performed (NCT04288713). Also corticosteroids are an option. Since we notice that some patients have persistent disease and at some point develop a proinflammatory profile especially a rise in CRP reflecting IL-6 elevation, which often leads to ICU admission this might be the timepoint to initiate potent anti-inflammatory therapy. For most patients this timepoint will be identified before the need of ICU admission and thus an anti-inflammatory drug might prevent them from ICU admission. This antiinflammatory strategy must be initiated together with the kallikrein-kinin pathway blockade and available antivirals as early as possible in disease. The anti-inflammatory strategies will buy time, but will not resolve the disease by themselves as long as the virus is present and or the bradykinin-induced angioedema is not resolved. A summary of these proposed targeted treatments and timing of treatment is depicted in *Figure 2*. An overview of the hypothesis on which this strategy is based is illustrated by *Figure 3A* (normal condition), 3B (mild inflammation) and 3C (severe inflammation).

## Conclusions

We are calling out for experts in the field of the kallikrein-kinin system and people involved in drug development to work together with SARS researchers who have the tools to test this hypothesis and interventions. This hypothesis explains the clinical spectrum that is so often observed and offers a rationale for treatment and more importantly timing of treatment. The bradykinin-driven pulmonary edema could be targeted by already available drugs such as icatibant or a plasma kallikrein inhibitor, such as lanadelumab. The cytokine-related clinical detoriation could respond to blocking the IL-1/IL-6 pathway. These treatment strategies, together with antiviral treatment, could prevent the development of ARDS in COVID-19 when started early and might be able to prevent ICU admission and the need for mechanical ventilation.

## Additional information

### Competing interests

Jos WM van der Meer: Senior editor, *eLife*. The other authors declare that no competing interests exist.

## Funding
No external funding was received for this work.

## Author contributions
Frank L van de Veerdonk, Roger J Brüggemann, Hans van der Hoeven, Conceptualization, Writing - original draft, Writing - review and editing; Mihai G Netea, Marcel van Deuren, Quirijn de Mast, Writing - original draft, Writing - review and editing; Jos WM van der Meer, Visualization, Writing - original draft, Writing - review and editing

## Author ORCIDs
Frank L van de Veerdonk (iD) https://orcid.org/0000-0002-1121-4894

## Decision letter and Author response
Decision letter https://doi.org/10.7554/eLife.57555.sa1
Author response https://doi.org/10.7554/eLife.57555.sa2

# Additional files

## Data availability
There are no datasets associated with this work.

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
