## [Decision Letter]

**Acceptance summary:**

After revision we believe that the manuscript has improved substantially and all comments of the reviewers have been answered in an appropriate manner. The potential target population is now clearly defined, and the theoretical benefit of kallikrein-kinin blockade in order to prevent progression of mild acute respiratory failure into severe ARDS in COVID-19 patients is well articulated. Needless to say, that despite the important message of the manuscript at present it remains a theory, hence we are looking forward to further clinical data on this topic.

**Decision letter after peer review:**

Thank you for submitting your article "Role of kinins and cytokines in the pathogenesis of COVID-19" for consideration by *eLife*. Your article has been reviewed by two peer reviewers, one of whom is a member of our Board of Reviewing Editors, and the evaluation has been overseen by Matthias Barton as the Senior Editor. The following individual involved in the review of your submission has agreed to reveal their identity: Eszter Nagy (Reviewer #2).

The reviewers have discussed the reviews with one another and the Reviewing Editor has drafted this decision to help you prepare a revised submission.

*Reviewer #1:*In this viewpoint the authors discuss their concept on the pathophysiology and the potential therapeutic approach acting via the kallikrein-kinin-bradykinin system in COVID-19 patients with acute respiratory failure. Their hypothesis is, that lung edema is mainly caused by angioedema via the bradykinin-dependent B1 and B2 receptors. They propose that blocking these receptors could attenuate the degree of angioedema hence improve patient outcomes.

Fighting COVID-19 is such a hot topic that is unprecedented in modern medicine. More than 200 scientific papers have been published daily for weeks. Finding solutions to ease the burden this pandemic has caused is one of the main duties for all of us. Therefore, the authors' aim is sound, their logic has strong pathophysiological rationale and their efforts should be congratulated. These are my concerns and comments.

1) The start their paper with the sentence: "COVID-19 patients present with pulmonary edema". Unfortunately, this is only true in one group of patients admitted with respiratory failure. One of the most interesting findings is that many of COVID-19 patients don't have pulmonary edema, in fact their pulmonary compliance and lung weight are normal despite the severe hypoxemia [Gattinoni et al., 2020]. Gattinoni et al., also recommends to divide the phenotypes into L and H types. The latter being the one in which pulmonary edema is present and a major concern. Therefore, I would suggest to change this sentence or delete it and also to consider this issue throughout the manuscript.

2) However, it doesn't mean that angioedema doesn't play an important role in the L phenotype, something that could also be elaborated in the manuscript. If this sounds too hypothetical, then I would only focus on patients in the H-type (High elastance – i.e.: low compliance; High right-to-left shunt; High lung weight; High recruitability).

3) Another important issue in the authors' concept could be the kidney. Although the paper is about the respiratory complications, but the title talks about COVID-19 in general. Those patients who go on to develop acute kidney injury (that is around 20% of the critically ill population in general, and 30% in those requiring mechanical ventilation), the mortality is around 80%! [ICNARC report 17 April 2020]. It has also been shown that ACE2 receptors can be found in the podocytes and the proximal tubule cells, therefore direct effect of the corona virus on the kidneys cannot be excluded [Pan X, et al., Intensive Care Med, 2020, 31 March]. In line with this, any successful therapy blocking this pathway may have beneficial effects not just on the lungs, but also on the kidneys. Something worth mentioning.

4) Regarding the conclusion. I don't entirely understand how detailed "instructions" on proning, ventilation settings happened to be in the conclusions. I would just simply state that lung protective measures are inevitable to avoid ventilator induced lung injury, if at all. Furthermore, I strongly disagree with the last sentence. One size does not fit all, and since thromboembolism is an extremely common feature in these patients, careful fluid management is highly recommended, and unnecessary hypovolemia, dehydration can potentially be lethal. I would in fact delete this whole lot and put the first sentence of the conclusion as last.*Reviewer #2:*The manuscript proposes the inhibition of the kallikrein-kinin-bradykinin system as a therapeutic approach for COVID-19. The authors outline clearly the rational and support their hypothesis and arguments with clinical observations and published data on SARS-CoV-2 or SASRS infections, as well as mechanistic overview of the kinin-bradykinin system at the molecular, cellular and tissue levels. The core hypothesis put forward is that by blocking the bradykinin pathway the bradykinin-induced local pulmonary angioedema can be prevented or reduced. The strength of the manuscript is that the authors also outline other treatment modalities and how to combine these in different stages of the disease.

1) Based on the role ACE2 in viral pathogenesis and the reduced ACE2 levels, the potential role of the kinin-bradykinin system in perivascular oedema in COVID-19 is convincing and bring this important aspect to the attention of field. The suggestion of blocking the bradykinin receptors is the most practical one, as a B2 blocker is available as a marketed drug. I suggest highlighting this, and be more pronounced that this should be considered as an immediate strategy and combine this drug with others that are being tested as repurposed drugs for COVID-19.

2) The section on the negative role of neutralizing anti-S-protein antibodies is very controversial. Unless it is proven otherwise with further research and immunological data, virus neutralizing anti-S antibodies are considered protective, at least for prophylaxis. Vast majority of the > 70 vaccine approaches against SARS-CoV-2 are based on the S-protein and the concept that such antibodies should be protective. The current passive immunization efforts (transfer of convalescent serum from patients who recovered from COVID-19) are also based on this notion. Antibody-mediated enhancement is only one of the potential explanations. A more balanced picture should be portrayed here and the conclusion tamed that the appearance of such antibodies is the cause of clinical worthening. It might be a co-incident, and the insufficient level of antibodies, quality or epitope specificity in certain patients might explain this observation. But again, experimental evidence (mainly with SARS) support immunization approaches based on the S-protein.

3) Obviously, the paper could be very much strengthened by experimental data, However, the high medical need to save lives in the current wave of the pandemic dismisses this requirement.

---

## [Author Response]

Reviewer #1:[…] 1) The start their paper with the sentence: "COVID-19 patients present with pulmonary edema". Unfortunately, this is only true in one group of patients admitted with respiratory failure. One of the most interesting findings is that many of COVID-19 patients don't have pulmonary edema, in fact their pulmonary compliance and lung weight are normal despite the severe hypoxemia [Gattinoni et al., 2020]. Gattinoni et al., also recommends to divide the phenotypes into L and H types. The latter being the one in which pulmonary edema is present and a major concern. Therefore, I would suggest to change this sentence or delete it and also to consider this issue throughout the manuscript.

We thank the reviewer for this nuance. We have adjusted the sentence by mentioning “can” present with and further focus on early disease in patients with COVID-19 where we think the pulmonary edema plays a major role in disease.

For the ICU we have incorporated the following text:

“In the ICU there are several striking observations. In contrast to patients with common forms of ARDS, approximately 70% of patients with severe COVID-19 show an only slightly decreased pulmonary compliance (L-type) (Gattinoni et al., 2020). Driving pressure is usually low. Recruitability is usually low and the use of high PEEP may therefore substantially increase functional residual capacity resulting in hyperinflation, high strain and considerable hypercapnia through an increase in dead space ventilation. Hereby mechanical ventilation may further contribute to lung damage. Only a minority of patients initially show the classical ARDS mechanical properties (H-type) with low compliance, high driving pressure and higher recruitability. Both L and H-type show high venous admixture. We and others have suggested that the L-type may progress to the H-type by a combination of negative intrathoracic pressure and increased lung permeability due to inflammation (so called patient-self inflicted lung injury P-SILI) (Gattinoni L et al., 2020).

2) However, it doesn't mean that angioedema doesn't play an important role in the L phenotype, something that could also be elaborated in the manuscript. If this sounds too hypothetical, then I would only focus on patients in the H-type (High elastance – i.e.: low compliance; High right-to-left shunt; High lung weight; High recruitability).

We agree, please see answer to comment 1.

3) Another important issue in the authors' concept could be the kidney. Although the paper is about the respiratory complications, but the title talks about COVID-19 in general. Those patients who go on to develop acute kidney injury (that is around 20% of the critically ill population in general, and 30% in those requiring mechanical ventilation), the mortality is around 80%! [ICNARC report 17 April 2020]. It has also been shown that ACE2 receptors can be found in the podocytes and the proximal tubule cells, therefore direct effect of the corona virus on the kidneys cannot be excluded [Pan X, et al., Intensive Care Med, 2020, 31 March]. In line with this, any successful therapy blocking this pathway may have beneficial effects not just on the lungs, but also on the kidneys. Something worth mentioning.

We have now added a sentence in the manuscript that targeting the kallikrein-kinin system might also have beneficial effects on other organs where a relative ACE2 deficiency due to virus could be present, such as in the gut and in the kidney.

4) Regarding the conclusion. I don't entirely understand how detailed "instructions" on proning, ventilation settings happened to be in the conclusions. I would just simply state that lung protective measures are inevitable to avoid ventilator induced lung injury, if at all. Furthermore, I strongly disagree with the last sentence. One size does not fit all, and since thromboembolism is an extremely common feature in these patients, careful fluid management is highly recommended, and unnecessary hypovolemia, dehydration can potentially be lethal. I would in fact delete this whole lot and put the first sentence of the conclusion as last.

We agree and have deleted this part from the manuscript.

Reviewer #2:[…] 1) Based on the role ACE2 in viral pathogenesis and the reduced ACE2 levels, the potential role of the kinin-bradykinin system in perivascular oedema in COVID-19 is convincing and bring this important aspect to the attention of field. The suggestion of blocking the bradykinin receptors is the most practical one, as a B2 blocker is available as a marketed drug. I suggest highlighting this, and be more pronounced that this should be considered as an immediate strategy and combine this drug with others that are being tested as repurposed drugs for COVID-19.

We thank the reviewer for this suggestion and we agree it should be highlighted more. We have adjusted this in the manuscript and have highlighted it in the Abstract and conclusions. We have also underscored this more in the last figure (Figure 3C) and discussed the rationale for targeting plasma kallikrein and B2 receptor more extensively.

2) The section on the negative role of neutralizing anti-S-protein antibodies is very controversial. Unless it is proven otherwise with further research and immunological data, virus neutralizing anti-S antibodies are considered protective, at least for prophylaxis. Vast majority of the > 70 vaccine approaches against SARS-CoV-2 are based on the S-protein and the concept that such antibodies should be protective. The current passive immunization efforts (transfer of convalescent serum from patients who recovered from COVID-19) are also based on this notion. Antibody-mediated enhancement is only one of the potential explanations. A more balanced picture should be portrayed here and the conclusion tamed that the appearance of such antibodies is the cause of clinical worthening. It might be a co-incident, and the insufficient level of antibodies, quality or epitope specificity in certain patients might explain this observation. But again, experimental evidence (mainly with SARS) support immunization approaches based on the S-protein.

We thank the reviewer for this important note. We decided to delete this from the hypothesis and focus on the predominant innate inflammation that accompanies the clinical worsening so this important message will stand and will not be distracted by something that is controversial and needs more discussion to put it in this manuscript.

3) Obviously, the paper could be very much strengthened by experimental data, However, the high medical need to save lives in the current wave of the pandemic dismisses this requirement.

We thank the reviewer for understanding. Studies with icatibant and lanadelumab are underway.